

# Recombinant hirudin attenuates pulmonary hypertension and thrombosis in acute pulmonary embolism rat model

Xiang Wei[1,2,*], Yanfen Zou[3,*], Shunli Dong[1,2], Yi Chen[1,2], Guoping Li[1,2] and Bin Wang[1,2]

[1] Department of Respiratory Medicine, Huzhou Central Hospital, Huzhou, Zhejiang Province, China
[2] Huzhou Key Laboratory of Precision Diagnosis and Treatment in Respiratory Diseases, Huzhou, Zhejiang Province, China
[3] Departments of Obstetrics and Gynecology, The Affiliated Yantai Yuhuangding Hospital of Qingdao University, Yantai, Shangdong Province, China
[*] These authors contributed equally to this work.

Corresponding authors
Guoping Li, lgp3allen@163.com

Bin Wang, wangbin@hzhospital.com

## ABSTRACT

**Background**. Acute pulmonary embolism (APE) is classified as a subset of diseases that are characterized by lung obstruction due to various types of emboli. Current clinical APE treatment using anticoagulants is frequently accompanied by high risk of bleeding complications. Recombinant hirudin (R-hirudin) has been found to have antithrombotic properties. However, the specific impact of R-hirudin on APE remains unknown.

**Methods**. Sprague-Dawley (SD) rats were randomly assigned to five groups, with thrombi injections to establish APE models. Control and APE group rats were subcutaneously injected with equal amounts of dimethyl sulfoxide (DMSO). The APE+R-hirudin low-dose, middle-dose, and high-dose groups received subcutaneous injections of hirudin at doses of 0.25 mg/kg, 0.5 mg/kg, and 1.0 mg/kg, respectively. Each group was subdivided into time points of 2 h, 6 h, 1 d, and 4 d, with five animals per point. Subsequently, all rats were euthanized, and serum and lung tissues were collected. Following the assessment of right ventricular pressure (RVP) and mean pulmonary artery pressure (mPAP), blood gas analysis, enzyme-linked immunosorbnent assay (ELISA), pulmonary artery vascular testing, hematoxylin-eosin (HE) staining, Terminal deoxynucleotidyl transferase-mediated dUTP-biotin nick end labeling (TUNEL) staining, immunohistochemistry, and Western blot experiments were conducted.

**Results**. R-hirudin treatment caused a significant reduction of mPAP, RVP, and Malondialdehyde (MDA) content, as well as $H_2O_2$ and myeloperoxidase (MPO) activity, while increasing pressure of oxygen ($PaO_2$) and Superoxide Dismutase (SOD) activity. R-hirudin also decreased wall area ratio and wall thickness to diameter ratio in APE rat pulmonary arteries. Serum levels of endothelin-1 (ET-1) and thromboxaneB2 (TXB2) decreased, while prostaglandin (6-K-PGF1$\alpha$) and NO levels increased. Moreover, R-hirudin ameliorated histopathological injuries and reduced apoptotic cells and Matrix metalloproteinase-9 (MMP9), vascular cell adhesion molecule-1 (VCAM-1), p-Extracellular signal-regulated kinase (ERK)1/2/ERK1/2, and p-P65/P65 expression in lung tissues.

**Conclusion**. R-hirudin attenuated pulmonary hypertension and thrombosis in APE rats, suggesting its potential as a novel treatment strategy for APE.

## INTRODUCTION

Pulmonary embolism has become the third leading cause of death from vascular disease, following cerebral infarction and heart attack (*Huang et al., 2023*). Acute pulmonary embolism (APE) often comes with high morbidity and mortality caused by endogenous or exogenous emboli obstructing the pulmonary arteries or blocking their branches (*Wu, Li & Tang, 2018*). When left untreated, it could cause irreversible pathophysiological changes such as chronic thromboembolic pulmonary hypertension and right heart failure, which alter the prognoses and quality of life of patients (*Ende-Verhaar et al., 2018*). Anticoagulants play a significant part in APE treatment by preventing thrombus expansion, embolism, and fresh thrombi formation and reducing mortality (*Lee, Davis & Kielly, 2016*). However, utilization of classical anticoagulant drug, warfarin, at standard effective dose also carried a high risk of bleeding and required attention to strict control of dosage, duration of administration and testing coagulation function of patients (*Alexander et al., 2021*). In addition, dosing form of heparin, intravenous or subcutaneous administration, may lead to compliance problems (*Alexander et al., 2021*). Therefore, finding better drugs for APE therapy is imperative. *Karpov et al. (2022)* pointed out that new approaches to studying improved prognosis in APE patients relied heavily on reliable animal models, including rat models. The construction of pulmonary artery autologous thrombus in Sprague-Dawley (SD) rats resulted in a rise in pulmonary artery resistance and pulmonary artery pressure within 24 h, which better mimicked changes in pulmonary artery pressure in APE humans during this period of time (*Karpov et al., 2022*). Therefore, we made selection to construct APE rat model to complete our study.

It was founded that APE mice developed decreased prostaglandin (6-K-PGF1$\alpha$) levels and increased thromboxaneB2 (TXB2) levels, and ameliorative effects of novel patented functional food, Mailiupian, in APE mice were accompanied by higher 6-K-PGF1$\alpha$ levels whereras lower TBX2 levels (*Shi et al., 2022*). In addition, significant oxidative stress occurred in APE patients, as evidenced by significantly elevated malondialdehyde (MDA), reactive oxygen species (ROS), and myeloperoxidase (MPO) levels (*Mühl et al., 2006*). Increased matrix metalloproteinase-9 (MMP9), p-extracellular signal-regulated kinase (ERK)1/2, and p-P65 protein expression were also found in APE cell and rat models (*Zhou et al., 2021*).

Hirudin, the active ingredient in the traditional Chinese medicine leech, has a broad spectrum of applications in cardiovascular and cerebrovascular system diseases due to its noteworthy anticoagulant and antithrombotic activities (*Dong et al., 2016*). As a direct thrombin inhibitor, different from heparin, hirudin exerts its anticoagulant effect

with suppression of thrombin-induced aggregation of platelets (*Wang et al., 2018b*). Currently, hirudin is primarily utilized for the clinical treatment of diffuse intravascular coagulation and cardiovascular diseases, prevention of deep vein thrombosis, prevention of thrombolysis or revascularization of thickened thrombus formation, and anticoagulation during haemodialysis (*Yu, Wang & Jin, 2018*). Recombinant hirudin (R-hirudin), the product of the most important active ingredient of leech obtained by genetic recombination technology, possesses essentially identical pharmacological activity and action as natural hirudin (*Junren et al., 2021*). R-hirudin has been found with ability to counteract thrombosis (*Cao & Li, 2018*). Furthermore, R-hirudin inhibited tumor growth, metastasis, and cell invasion by suppressing MMP9 expression, thereby inhibiting progression of non-small cell lung cancer, whose acute complications typically included APE (*Zhao et al., 2022a*). Inhibition of MDA levels with R-hirudin treatment occurred in diabetic cataract rats (*Gong, Zhang & Tan, 2013*). Recently, *Men et al., (2022)* demonstrated that R-hirudin microneedle patches, a transdermal delivery modality, exhibited anticoagulant effects and were capable of preventing APE development. However, therapeutic effects and specific mechanism of R-hirudin on APE remain insufficiently explored.

Consequently, based on these above findings, we made a reasonable speculation that R-hirudin was able to inhibit platelet aggregation and improve vascular endothelial function by regulating levels of oxidative stress and vasoactive factor expression, thereby alleviating APE symptoms. This investigation focused on examining effects of R-hirudin on APE model rats with thrombosis, thereby providing a theoretical basis for widespread utilization of R-hirudin in the clinical treatment of APE.

# MATERIALS AND METHODS

## Animals

We purchased 100 male SD rats ($250 \pm 20$ g) from Shanghai SLAC Laboratory Animal Co., Ltd., SCXK (Hu) 2022-0004 and housed them at Zhejiang Eyong Pharmaceutical Research and Development Center (SYXK (Zhe) 2021-0033). The rats were housed individually and kept under standard conditions with ambient temperature ($20-22\ °C$), relative humidity (45–50%), and a day-and-night-cycle of 12 h light and 12 h dark. All rats had free access to food and water. One week of acclimatization feeding was completed prior to experiments. They were euthanized before the end of experiment program. Details of the euthanasia method used are described below. In clean cages, rats were euthanized *via* inhalation of compressed $CO_2$ gas in cylinders (flow rate: 50%–60% of cage volume/min, no prefilled chambers). No other animals were present at the scene of execution, and the carcasses were not properly disposed of until the rat was confirmed dead. All animal experiments were conducted in accordance with the Guide for the Care and Use of Laboratory Animals. All animal tests were approved by the Laboratory Animal Management and Welfare Ethical Review Committee of Zhejiang Eyong Pharmaceutical Research and Development Center (approval no. ZJEY-20230131-03).
## APE rat models and grouping

Blood was collected from orbital vein of rats under aseptic conditions, left to coagulate for 4 h to make a thrombus embolus (1.1 mm × 2.0 mm), and then added 2 mL of physiological saline to produce thrombus suspension, refrigerated at 4 °C. The rats were anesthetized with 2% sodium pentobarbital (50 mg/kg) intraperitoneally. They were placed in a supine position and the left side of the neck was trimmed and disinfected. After longitudinal incision and isolation of external jugular vein, a catheter needle was then inserted into the pulmonary artery and approximately 30 thrombi were generated by injecting 2 mL physiological saline slowly and uniformly. The incision was then sutured and the APE model was established (*Zhang et al., 2017*; *Wang et al., 2014*). When the rats showed obvious cyanosis and accelerated and deepened respiration, the modeling was considered successful. All rats were successfully modeled. No rats experienced adverse events such as rapid death due to large pulmonary embolism or sudden weight loss during modeling. If any of the above situations occurred, to minimize suffering of animals, we were prepared to promptly terminate the experiment and euthanize the rats.

Based on previous research and the 4R principles and statistics of experimental animals, SD rats were randomly partitioned into five groups utilizing a random number table method (each group was subdivided into 2 h, 6 h, 1 d, and 4 d time points, with five animals at each time point; $n = 5$): Control, APE, APE+R-hirudin low-dose (Low), APE+R-hirudin middle-dose (Medium), and APE+R-hirudin high-dose (High). Rats in the control and the APE group were injected subcutaneously with equivalent quantities of dimethyl sulfoxide (DMSO). In accordance with previous published applications and studies of R-hirudin in rats, we ultimately settled on 0.25 mg/kg, 0.5 mg/kg, and 1.0 mg/kg as R-hirudin (RE120A, Hyphen-BioMed, France) doses for Low, Medium, and High groups, respectively (*Jiang et al., 2013*; *Just, Tripier & Seiffge, 1991*). All groups were dosed by subcutaneous injection 2 h prior to modeling.

## Right ventricular pressure (RVP) and mean pulmonary artery pressure (mPAP) assessment

Rats in each group were anesthetized *via* intraperitoneal administration of 2% pentobarbital sodium at 2 h, 6 h, 1 d, and 4 d time points after thrombus insertion (with five animals at each time point). The right external jugular vein was isolated and a polyphenylene tube (0.9 mm) was inserted into the right atrium, right ventricle, and pulmonary artery with an internal diameter, and RVP and mPAP were measured and recorded.

## Blood gas analysis

Rats were cannulated using the carotid artery before execution and arterial blood was placed on a fully automated blood gas analysis machine for blood gas analysis. Changes in partial pressure of oxygen ($PaO_2$) and partial pressure of carbon dioxide ($PaCO_2$) in rats were detected.

## Enzyme-linked immunosorbnent assay (ELISA) assay

Following euthanasia by $CO_2$, rat left lung tissues were isolated, tissue homogenate was carefully obtained, supernatant was taken, and the ELISA kits (Shanghai Enzyme

Link Biotechnology Co., Ltd., Shanghai, China) were employed to assess MDA content, Superoxide Dismutase (SOD), $H_2O_2$, and MPO activity. The levels of Endothelin-1 (ET-1), TXB2, 6-K-PGF1$\alpha$, and NO in serum were detected using ELISA kits (Shanghai Enzyme Link Biotechnology Co., Ltd.).

## Pulmonary artery vascular testing

Five cross-sections of the pulmonary arteries were taken from each rat. The CMIAS2001 B multifunctional true colour pathological image analysis system was utilized. To ensure that size of pulmonary arteries did not affect comparison, wall area ratio and wall thickness to diameter ratio were selected as statistical indicators to detect intimal hyperplasia in pulmonary arteries.

## Hematoxylin-eosin (HE) staining

Rat lung tissues were fixed with 4% paraformaldehyde, embedded in paraffin, and sectioned. Subsequently, HE staining was performed to observe pathological changes, and photographs were taken. Data were analyzed by an observer who was blinded to treatment groups.

## Terminal deoxynucleotidyl transferase-mediated dUTP-biotin Nick End Labeling (TUNEL) staining

Apoptosis of lung tissue cell was detected by TUNEL kit (G1501; Servicebio, Wuhan, China). The lung tissues were fixed with 4% paraformaldehyde, treated with 0.1% Triton X-100, and then incubated with 50 µL TUNEL reaction mixture. The lung tissue cells were subsequently counterstained with DAPI (G1012; Servicebio) and slices were observed under a microscope (Eclipse C1; Nikon, Tokyo, Japan).

## Immunohistochemistry

Paraffin sections of lung tissue were first dewaxed to water, and then antigenic repair was performed to block endogenous peroxidase. After serum sealing, primary antibodies MMP9 antibody (AF5228; Affinity, Cincinnati, OH, USA), and vascular cell adhesion molecule-1 (VCAM-1) antibody (DF6082; Affinity) were added. Subsequently, horseradish secondary rabbit peroxidase antibody (HRP) (AF5131; Abcam, Cambridge, UK) and 50 mM Tris–HCl buffer (pH 7.4) were added and incubated. It was further exposed to DAB solution (G1212-200T; Servicebio) for 15 min and reacted with hematoxylin for 3 min.

## Western blot

The total protein in lung tissue was collected and total protein concentration was detected utilizing the BCA protein kit (pc0020; Solarbio). The PVDF membrane was blocked with 5% skimmed milk powder, and the primary antibodies ERK1/2 (4695T; CST, Danvers, MA, USA), p-ERK1/2 (4370T; CST), P65 (AF5006; Affinity), and p-P65 (AF2006; Affinity) were added and incubated overnight. The membrane was then rinsed and HRP incubated. The protein bands were detected by the ECL and protein gray value was calculated by Image J.

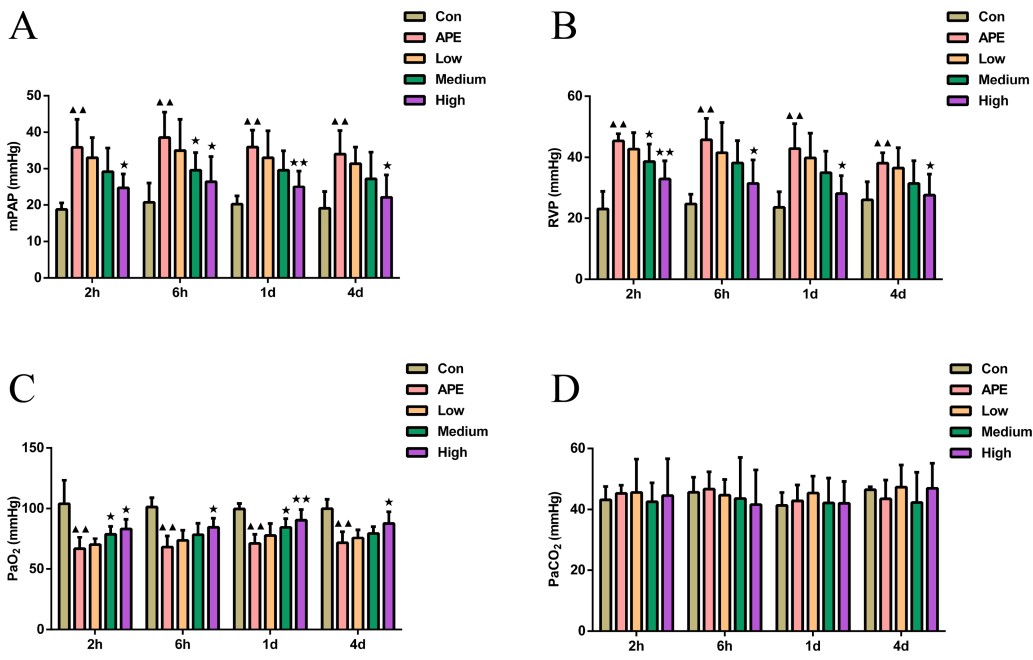

**Figure 1** **R-hirudin improved cardiorespiratory function and blood gas indices with reduced mPAP and RVP and increased PaO₂ in APE rats.** Determination of mPAP (A), RVP (B), PaO$_2$ (C), and PaCO$_2$ (D) was performed in all rats. Data were expressed as mean ± SD, $n = 5$. Compared to the Con group, ▲▲$P < 0.01$. Compared to the APE group, *$P < 0.05$, **$P < 0.01$. Con: control, APE: acute pulmonary embolism, Low: APE+R-hirudin low dose, Medium: APE+R-hirudin middle dose, High: APE+R-hirudin high dose, RVP: right ventricular pressure, mPAP: mean pulmonary artery pressure, PaO$_2$: arterial blood oxygen pressure, PaCO$_2$: partial pressure of carbon dioxide.

## Statistical analysis

We used SPSS 21.0 to analyze data, one-way ANOVA for multiple groups, and subsequent Tukey's test for intergroup comparisons. The Kruskal-Wallis H test was applied to test for variance heterogeneity. Data were expressed as mean ± standard deviation (mean ± SD), and $P < 0.05$ indicated that difference was statistically significant.

## RESULTS

### R-hirudin improved cardiorespiratory function and blood gas indices with reduced mPAP and RVP and increased PaO2 in APE rats

As shown in Figs. 1A–1B, mPAP and RVP in APE group were higher at 2 h, 6 h, 1 d, and 2 d compared to control mice ($P < 0.01$). In comparison with APE mice, mPAP at 6 h and RVP at 2 h were lower in R-hirudin medium dose group ($P < 0.05$), and mPAP and RVP at 2 h, 6 h, 1 d, and 2 d were lower in R-hirudin high dose group ($P < 0.05$). Figure 1C showed that arterial PaO$_2$ was lower in APE group than that of control mice at 2 h, 6 h, 1 d, and 2 d ($P < 0.01$). Compared to APE group, arterial PaO$_2$ was augmented at 2 h and 1 d in R-hirudin medium dose group ($P < 0.05$), and at 2 h, 6 h, 1 d, and 2 d in R-hirudin high dose group ($P < 0.05$). However, R-hirudin had no significant effect on PaCO$_2$ (Fig. 1D).

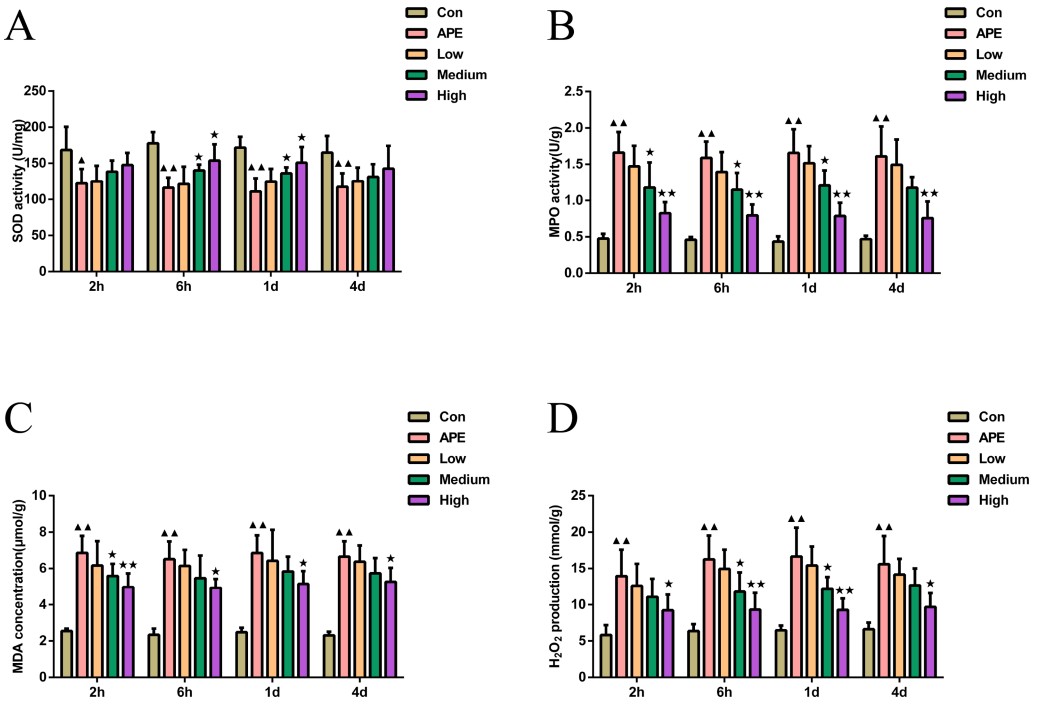

**Figure 2** **R-hirudin ameliorated the level of oxidative stress with reduced MPO, MDA, and $H_2O_2$ activity whereas increased SOD activity in lung tissues of APE rats.** Detection of SOD (A), MPO (B), MDA (C), and $H_2O_2$ (D) was done in all rats by ELISA assay. Data were expressed as mean $\pm$ SD, $n = 5$. Compared to the Con group, $^{\blacktriangle}P < 0.05$, $^{\blacktriangle\blacktriangle}P < 0.01$. Compared to the APE group, $^{*}P < 0.05$, $^{**}P < 0.01$. Con: control, APE: acute pulmonary embolism, Low: APE+R-hirudin low dose, Medium: APE+R-hirudin middle dose, High: APE+R-hirudin high dose.

## R-hirudin ameliorated the level of oxidative stress with reduced MPO, MDA, and $H_2O_2$ activity whereas increased SOD activity in lung tissues of APE rats

As demonstrated in Fig. 2, MDA, $H_2O_2$ levels, and MPO activity at 2 h, 6 h, 1 d, and 2 d were found higher in APE group than control mice ($P < 0.01$), but activity of SOD was lower ($P < 0.05$). Compared with APE group, R-hirudin medium dose group exhibited lower $H_2O_2$ levels at 6 h and 1 d, as well as lower activity of MPO at 2 h, 6 h, 1 d, and MDA at 2 h ($P < 0.05$). However levels of SOD at 6 h and 1 d were higher ($P < 0.05$). The levels of MDA, $H_2O_2$, and MPO activity were significantly decreased at 2 h, 6 h, 1 d, and 2 d in R-hirudin high dose group ($P < 0.05$), but SOD activity increased at 6 h and 1 d ($P < 0.05$).

## R-hirudin enhanced vascular endothelial function in APE rats

As shown in Figs. 3A–3B, the wall area ratio of the pulmonary artery in the APE group was higher at 4 d ($P < 0.01$) compared to the control group, and the wall thickness to diameter ratio was higher at 1 d and 4 d ($P < 0.01$). In comparison with APE group, the wall area ratio of pulmonary arteries in R-hirudin medium dose group exhibited significant

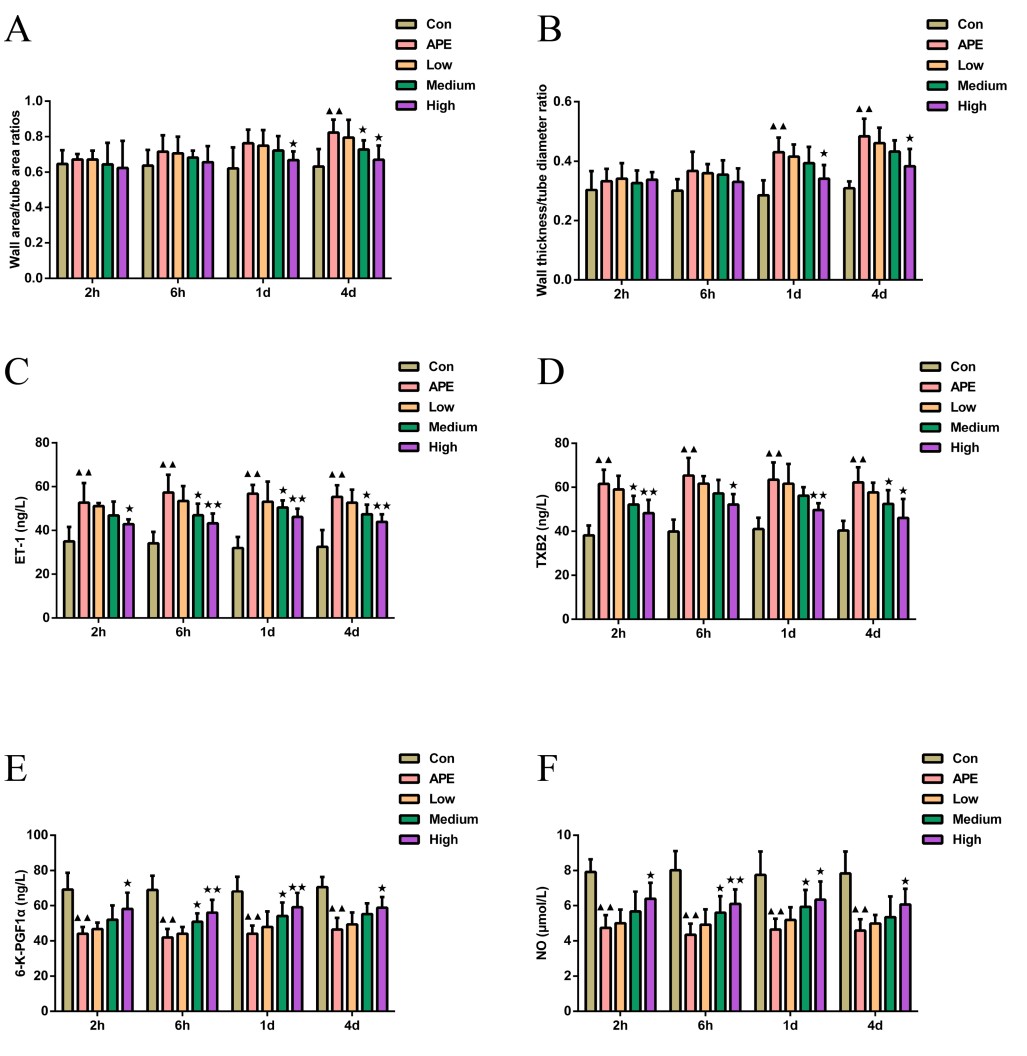

**Figure 3  R-hirudin enhanced vascular endothelial function in APE rats.** (A–B) Examination of the wall area ratio and wall thickness to diameter ratio of pulmonary arteries was made in all rats; determination of ET-1 (C), TXB2 (D), 6-K-PGF1 $\alpha$ (E), and NO (F) was conducted in all rats by ELISA assay. Data were expressed as mean ±SD, $n = 5$. Compared to the Con group, ▲▲$P < 0.01$. Compared to the APE group, *$P < 0.05$. **$P < 0.01$. Con: control, APE: acute pulmonary embolism, Low: APE+R-hirudin low dose, Medium: APE+R-hirudin middle dose, High: APE+R-hirudin high dose.

reduction at 4 d ($P < 0.05$), and the wall area ratio and wall thickness to diameter ratio of pulmonary arteries in the High group were lower at 1 d and 4 d ($P < 0.05$).

As illustrated in Figs. 3C–3F, the serum levels of ET-1 and TXB2 were higher ($P < 0.01$), while the levels of 6-K-PGF1$\alpha$ and NO were lower in the APE group at 2 h, 6 h, 1 d, and 2 d ($P < 0.01$) compared to the control group. When comparing the APE group to the R-hirudin intermediate dose group, serum levels of ET-1 at 6 h, 1 d, and 4 d and TXB2 at 6 h and 1d were lower ($P < 0.05$), although the levels of 6-K-PGF1$\alpha$ and NO at 6 h and 1 d increased ($P < 0.05$). In the R-hirudin high dose group, serum levels of ET-1 and TXB2

| Con | APE | Low | Medium | High |

**Figure 4** **R-hirudin ameliorated histopathological changes in lung tissues of APE rats (400 ×, bar = 50 µm).** The histopathological changes were detected by HE staining. Con: control, APE: acute pulmonary embolism, Low: APE+R-hirudin low dose, Medium: APE+R-hirudin middle dose, High: APE+R-hirudin high dose.

at 2 h, 6 h, 1 d, and 2 d were reduced ($P < 0.05$), while 6-K-PGF1$\alpha$ and NO levels were markedly elevated ($P < 0.05$).

## R-hirudin ameliorated histopathological changes in lung tissues of APE rats

Figure 4 revealed that the histopathological sections of lungs in the control group showed normal intact bronchial epithelium with a small amount of inflammatory cell infiltration around the local blood vessels and no obvious thrombus. In the APE group, the bronchial epithelium in the lung tissue was detached, and thrombus was evident in the lumen. Additionally, there was a large number of inflammatory cells infiltrating the lumen, while only a small number of inflammatory cells infiltrated around the blood vessels. In the Low and Medium groups, a few bronchial epithelial cells were observed in the lung tissue; with inflammatory cell infiltration, a small number of leukocyte aggregates and thrombi were seen in the local vascular lumen. Conversely, almost normal bronchial epithelium was visible and intact in the lung tissue of the High group.

## R-hirudin reduced cell apoptosis in lung tissues of APE rats

According to Fig. 5, the rate of positive cells examined using TUNEL staining assay in the APE group was greater compared to control group ($P < 0.01$). In contrast to the APE group, apoptosis in the Medium and High groups occurred decreased ($P < 0.01$).

## R-hirudin decreased expression of MMP9 and VCAM-1 in lung tissues of APE rats

As shown in Fig. 6, MMP9 and VCAM-1 expression detceted by immunohistochemistry assay in rat lung tissues increased in the APE group *versus* the control group ($P < 0.01$). When compared to the APE group, the expression of MMP9 in the lung tissues of rats in the R-hirudin low, medium, and high dose groups was lower (R-hirudin low and medium dose groups: $P < 0.05$; R-hirudin high dose group: $P < 0.01$), and VCAM-1 expression in the lung tissues of rats in the R-hirudin medium and high dose groups was reduced (R-hirudin medium dose group: $P < 0.05$; R-hirudin high dose group: $P < 0.01$).

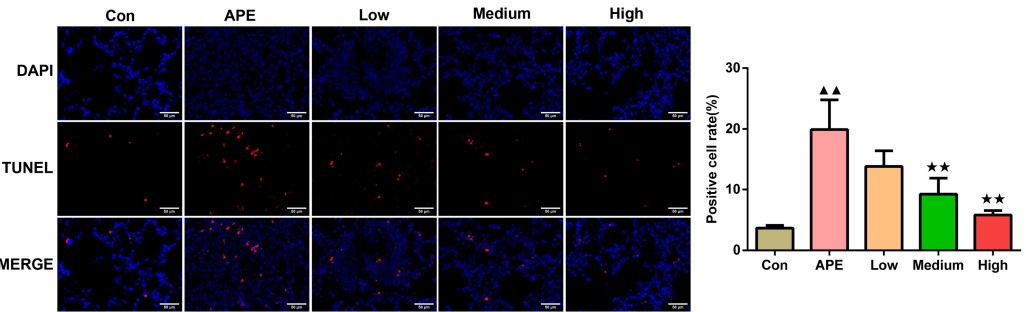

**Figure 5** **R-hirudin reduced cell apoptosis in lung tissues of APE rats (200 ×, bar = 50 μm).** The apoptosis was determined using TUNEL staining. Data were expressed as mean ±SD, n = 3. Compared to the Con group, ▲▲P < 0.01. Compared to the APE group, **P < 0.01. Con: control, APE: acute pulmonary embolism, Low: APE+R-hirudin low dose, Medium: APE+R-hirudin middle dose, High: APE+R-hirudin high dose.

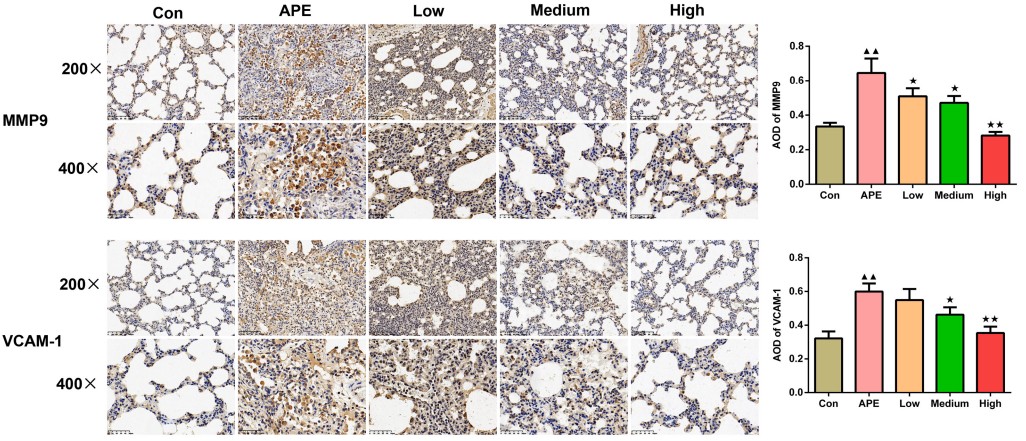

**Figure 6** **R-hirudin decreased expression of MMP9 and VCAM-1 in lung tissues of APE rats (200 ×, bar =100 μm; 400 ×, bar = 50 μm).** The MMP9 and VCAM-1 expression was examined by immunohistochemistry. Data were expressed as mean ±SD, n = 3. Compared to the Con group, ▲▲P < 0.01. Compared to the APE group, *P < 0.05, **P < 0.01. Con: control, APE: acute pulmonary embolism, Low: APE+R-hirudin low dose, Medium: APE+R-hirudin middle dose, High: APE+R-hirudin high dose.

## R-hirudin reduced the protein expression of p-ERK1/2/ERK1/2 and p-P65/P65 in lung tissues of APE rats

Figure 7 showed that when compared to control group, p-ERK1/2/ERK1/2 and p-P65/P65 levels determined by Western blot assay were elevated in lung tissues of APE, R-hirudin low-dose, and R-hirudin middle-dose groups (APE group and R-hirudin low-dose group: P < 0.01; R-hirudin middle-dose group: P < 0.05). In contrast to the APE group, p-ERK1/2/ERK1/2 and p-P65/P65 protein expression was lower in the lung tissues of the Medium and High groups (P < 0.01).

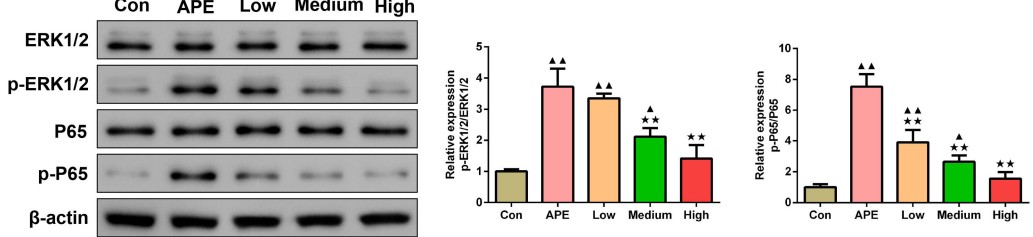

**Figure 7  R-hirudin reduced the protein expression of p-ERK1/2/ERK1/2 and p-P65/P65 in lung tissues of APE rats.** The p-ERK1/2/ERK1/2 and p-P65/P65 protein expression was detected by Western blot assay. Data were expressed as mean ± SD, $n = 3$. Compared to the Con group, $^{\blacktriangle}P < 0.05$, $^{\blacktriangle\blacktriangle}P < 0.01$. Compared to the APE group, $^{**}P < 0.01$. Con: control, APE: acute pulmonary embolism, Low: APE+R-hirudin low dose, Medium: APE+R-hirudin middle dose, High: APE+R-hirudin high dose.

## DISCUSSION

APE, one of the leading causes of sudden death in common acute and critical conditions, poses a great threat to human health (*He et al., 2022*). R-hirudin exhibits a crucial anti-thrombotic effect (*Hassan et al., 2008*). In this study, R-hirudin was found to improve pulmonary hypertension and thrombosis in rats, and our results provided a novel perspective on APE treatment.

Pulmonary hypertension can act as a good indicator of right heart insufficiency in APE (*Alonso-Martínez et al., 2015*). Blood gas analysis, both $PaO_2$ and $PaCO_2$ values, can be taken as clinical criteria for detecting APE, essential indicators for determining degree of hypoxia in patients, of which APE patients are often accompanied by hypoxemia, with lowered $PaO_2$ values as main symptom (*Becattini et al., 2018*). The results of *Zhou et al. (2017)* confirmed that mPAP of APE rats showed higher than that of sham-operated rats, whereas no significant difference in RVP. Furthermore, decreased $PaO_2$ and higher blood pressure in APE patients were also demonstrated in an observational study, and the effects were reversed after drug treatment (*Wang et al., 2018a*). These above studies further illustrated reliability of our findings. In this study, it was observed that R-hirudin intervention reduced mPAP and RVP, but increased $PaO_2$ in APE rats, which exerted a protective effect on pulmonary respiratory function. We noted that there were no significant changes in $PaCO_2$ between groups, suggesting that R-hirudin intervention did not cause severe $CO_2$ retention. Rats with chronic obstructive pulmonary disease had a larger tubular wall area and higher wall thickness in both fine bronchi and muscular arteries than in normal rats (*Wang et al., 2017*), which was similar to higher pulmonary artery wall area ratio and wall thickness to tube diameter ratio in APE rats observed in this study. The present study further implied that R-hirudin could reverse these changes and ameliorate lung tissue destruction.

MPO, MDA, and SOD represent significant indicators of the level of oxidative stress in lung tissues (*Zhao et al., 2022b*). *Yu, Wang & Jin (2018)* identified that hirudin inhibited MDA content and enhanced SOD activity, further alleviating Ang II-induced oxidative stress, which was in agreement with our findings. NO plays a pivotal role in

the regulation of development of pulmonary hypertension as an endothelium-derived diastolic factor (*Antwi-Boasiako & Campbell, 2018*). In addition, NO exerts vasodilatory and inhibitory effects on vascular cell proliferation and migration and acts as an important regulator of the maintenance of vascular wall tone in pulmonary circulation and prevention of pulmonary vascular remodeling (*Hagan & Pepke-Zaba, 2011*). The reverse regulator of NO is ET-1, a vasoconstrictor active substance synthesized and secreted mainly by vascular endothelial cells (*Thengchaisri et al., 2015*). In the present analysis, APE rats displayed markedly reduced NO levels and enhanced ET-1 levels, and a significant increase in NO levels and a decrease in ET-1 occurred after R-hirudin intervention, demonstrating that R-hirudin, in addition to ameliorating oxidative stress in APE rats, also led to improvement in vascular endothelial function, , thereby resisting APE. Thromboxane A2 (TXA2) exhibits a strong vasoconstrictive effect and induces platelet aggregation, causing microthrombosis (*Liu et al., 2017*). However, since TXA2 is unstable and its metabolite TXB2 is more stable, plasma TXB2 levels are frequently utilized to evaluate thromboxane activity (*Yang et al., 2022*). 6-K-PGF1$\alpha$ is a stable metabolite of prostacyclin, which inhibits platelet activation and dilates blood vessels effects (*Ma et al., 2018*). It was founded elevation of TXA2 and ET-1 in APE rabbit model (*Zhang et al., 2018*). In this study, R-hirudin decreased ET-1 and TXB2 levels, and increased 6-K-PGF1$\alpha$ and NO levels. This was similar to the findings of *Abulizi et al. (2023)*, who identified increased serum NO and 6-K-PGF1$\alpha$ levels, decreased ET-1 and TXB2 levels in rats with atherosclerosis, resulting in anti-thrombotic outcomes. Therefore, we made the hypothesis that R-hirudin attenuated oxidative stress and inflammatory response, suppressed platelet aggregation, and promoted restoration of vascular endothelial function in APE rats, which in turn exerted anti-thrombotic effects and protective effects on lung tissues.

VCAM-1 is an inflammatory mediator that promotes the development of lung fibrosis (*Vallejo et al., 2018*). MMP-9 is a crucial member of the matrix metalloproteinase family, which is implicated in airway remodeling and induces mesenchymal cell proliferation and alveolar epithelial damage, resulting in lung function impairment (*Pardo et al., 2016*). *Zhou et al. (2021)* found increased MMP-9 protein expression in cardiomyocytes of rats with pulmonary embolism, similar to our results. Additionally, a recent study revealed increased VCAM-1-mediated vascular inflammation in a mouse model of acute lung diseases (*Kim et al., 2023*). Hirudin inhibited up-regulation of VCAM, thereby inhibiting thrombus formation (*Chen et al., 2001*). In this investigation, we detected that R-hirudin reduced MMP9 and VCAM-1 expression in the lung tissues of APE rats, along with attenuation of inflammatory cell infiltration and apoptosis inhibition, further supporting that R-hirudin may improve APE by inhibiting MMP9 and VCAM-1 and suppressing inflammatory response.

ERK1/2 constitutes an integral component of the MAPK family, and numerous studies have demonstrated that ERK1/2 signaling pathways are implicated in the regulation of airway inflammation and airway smooth muscle proliferation, and act as common signaling pathways (*Zhang et al., 2022*; *Goldklang et al., 2013*). P65 could regulate the transcription of various genes involved in the immune response, including cytokines and inflammatory mediators (*Zhao et al., 2020*). Expression of p-ERK1/2 and p-P65/P65

was founded elevated in cardiomyocytes of rats with pulmonary embolism (*Zhou et al., 2021*), consistent with our findings. In our study, R-hirudin attenuated p-ERK1/2/ERK1/2 and p-P65/P65 expression in lung tissues of APE rats in a dose-dependent manner. This aligned with the findings from previous research in which the expression of p-ERK1/2/ERK1/2 and p-P65/P65 was alleviated and lung fibrosis was attenuated (*Yao et al., 2021*). We reasonably inferred from these results that R-hirudin may ameliorate APE symptoms by down-regulating p-ERK1/2/ERK1/2 and p-P65/P65 expression in APE rat lung tissues through a dose-dependent process.

In this study, we established APE rat models with rat autologous pulmonary thromboembolism assay. A modified trans-jugular vein embolization method with stable emboli was used to ensure homogeneity of models. Although as a representative APE model, it still has certain limitations. The SD rats have a highly efficient endogenous fibrinolysis system, and autologous blood clots do not produce permanently fixed pulmonary embolism model (*Karpov et al., 2022*). Approximately 95% lysis of rat pulmonary artery autologous thrombi occurs within 24 h (*Karpov et al., 2022*). Moreover, the blood that can be extracted from rats is limited, so this process need to be further improved. It is also necessary for development of a durable fixed thrombus model that is more compatible with the pathophysiological process of clinical acute pulmonary thromboembolism in future studies.

## CONCLUSION

In conclusion, we were able to infer that R-hirudin treatment may be able to attenuate pulmonary arterial hypertension and thrombosis, improve APE, and protect lung tissues by improving cardiopulmonary function and vascular endothelial function, alleviating oxidative stress injury and inflammatory response, dose-dependently regulating MMP-9, VCAM-1, p-ERK1/2/ERK1/2 and p-P65/P65 expression, and inhibiting apoptosis. R-hirudin may serve as a promising therapy for APE, providing new options and research directions for APE treatment. In future studies, we may further verify molecular mechanism of the protective effects of R-hirudin on APE, with aim of providing more valuable and stronger evidence to support future clinical applications. Moreover, since complex pathogenesis of APE, whether R-hirudin may ameliorate APE through other pathways still requires further research and exploration.

### Funding
This study was supported by The Public Technology Applied Research Program of Huzhou City (Program no. 2022GY05). The funders had no role in study design, data collection and analysis, decision to publish, or preparation of the manuscript.

### Grant Disclosures
The following grant information was disclosed by the authors:
The Public Technology Applied Research Program of Huzhou City: 2022GY05.

## Competing Interests

The authors declare there are no competing interests.

## Author Contributions

- Xiang Wei performed the experiments, analyzed the data, prepared figures and/or tables, authored or reviewed drafts of the article, and approved the final draft.
- Yanfen Zou performed the experiments, analyzed the data, prepared figures and/or tables, authored or reviewed drafts of the article, and approved the final draft.
- Shunli Dong analyzed the data, prepared figures and/or tables, authored or reviewed drafts of the article, and approved the final draft.
- Yi Chen conceived and designed the experiments, prepared figures and/or tables, and approved the final draft.
- Guoping Li performed the experiments, analyzed the data, prepared figures and/or tables, and approved the final draft.
- Bin Wang conceived and designed the experiments, prepared figures and/or tables, and approved the final draft.

## Animal Ethics

The following information was supplied relating to ethical approvals (*i.e.*, approving body and any reference numbers):

Laboratory Animal Management and Welfare Ethical Review Committee of Zhejiang Eyong Pharmaceutical Research and Development Center (approve no. ZJEY-20230131-03)

## Data Availability

The data is available in the Supplemental File.

## Supplemental Information

Supplemental information for this article can be found online at http://dx.doi.org/10.7717/peerj.17039#supplemental-information.

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
