# Peer review of "Recombinant hirudin attenuates pulmonary hypertension and thrombosis in acute pulmonary embolism rat model"

_PeerJ, doi:10.7717/peerj.17039_

## Round 0.1 · original submission · Major Revisions

Please carefully read the comments from all reviewers and provide a point-by-point response accordingly.

·

Basic reporting

In this manuscript, authors describe effects of hirudin on pulmonary hypertension and thrombosis in acute pulmonary embolism. The manuscript is written well. My primary concern in design of animal experiments, was power analysis done before they used 5 mice each group? Figure 3, left 2 graphs, the axis legends are not clear. Lastly, some places very wage and indirect words have been used, like novel prospective word.

Experimental design

no comment

Validity of the findings

no comments

Additional comments

The data looks suppers doubt.

Reviewer 2 ·

Basic reporting

Abstract:
Initially, the first occurrence of an English abbreviation should be accompanied by its full English name. Please check it.
The background information should more directly establish the current gap in APE treatment that this study addresses.
Introduction:
A more detailed explanation of the mechanism of action of R-hirudin and its relevance to APE could be elucidated.
The introduction could benefit from a clearer statement of the study's objectives or hypotheses.
The literature review should critically evaluate existing studies, highlighting how this study adds new knowledge.
Potential limitations and side effects of current APE treatments that this study aims to overcome should be discussed.
The rationale for using a rat model and its relevance to human APE should be more thoroughly justified.

Experimental design

Materials and Methods:
The selection criteria for animal models and the reasons for choosing specific doses of R-hirudin should be detailed.
A more comprehensive description of the control measures and experimental conditions is needed.
Results and Discussion:
A critical evaluation of any unexpected or inconsistent findings is needed.
The discussion should provide a more in-depth analysis of how the findings relate to and expand upon existing literature.
Potential mechanisms underlying the observed effects should be explored.
Limitations of the study, such as the extrapolation of results from rats to humans, should be critically discussed.
The implications of these findings for future APE research and potential treatment strategies should be emphasized.
The discussion could propose specific future research directions based on the study's outcomes.

Validity of the findings

Figures and Tables:
Figure1: The graph on the left is the ordinate mPAP? It is not noted on the Figure caption. In addition, ▲P<0.05 is not presented in the figure, please check them carefully.
In Figure5 (6-k-PGF1a), the caption (control, APE, low, Medium, High) in the figure is inconsistent to others, please change the style for consistency.
Please rewrite the Figure caption. Remember writing effective figure captions in scientific manuscripts involves several key requirements: Captions should be brief yet sufficiently descriptive to understand the figure without referring to the text. Use clear and precise language. The caption should explain what the figure depicts and its relevance to the study. Include necessary details such as what is being shown, the methodology used (e.g., type of assay, statistical analysis), and any relevant conditions or specific aspects highlighted in the figure. If applicable, briefly state the main findings shown in the figure. However, avoid extensive interpretation or discussion of the results. Explain all symbols, lines, bars, or other elements used in the figure. If the figure contains multiple panels or parts, label each clearly (e.g., A, B, C) and describe them individually. The caption should link the figure to the relevant text in the manuscript, guiding the reader on where to find more detailed discussion or context. Ensure the caption is understandable to a broad audience, including those who might not be specialists in the specific field. Avoid repeating information already stated in the text or other figures.

Additional comments

The manuscript should be proofread for grammatical and typographical errors.

Reviewer 3 ·

Basic reporting

no comment

Experimental design

no comment

Validity of the findings

no comment

Additional comments

1. Please replenish purchase sources or information of R-hirudin, including manufacturer and article number.
2. In line 59 of this manuscript, it was mentioned that "However, prolonged usage of anticoagulants may lead to associated side effects such as bleeding risks and complications". In fact, several studies have mentioned that although R-hirudin has been a new type of anti-coagulant and anti-thrombotic drug, there are certain bleeding risks and complications associated with the current clinical use of R-hirudin. So what is the significance and advantages of this study?
3. It is recommended enhancement for the relevance of this study in the "Introduction" section. Acute complications of lung cancer typically include the development of pulmonary embolism. A recent publication in journal BJP in 2022 found that R-hirudin could inhibit tumor growth and metastasis and cell invasion by suppressing the expression of MMP9 and IL6, and thus inhibit progression of non-small cell lung cancer.
4. Please clarify the specific reference or source or reasons for selection of low, medium and high doses of R-hirudin.
5. In lines 248 and 251 of "Results" section, please correct "p-ERK1/2 and p-P65 protein levels" to proper protein expression, like "expression of p-ERK1/2/ERK1/2 and p-P65/P65" mentioned in the subheadings and "Discussion" section.
6. TUNEL staining to determine apoptosis was not mentioned in "Discussion" section, what was the purpose of this experiment? If it was done to fully demonstrate that R-hirudin could inhibit apoptosis, apoptosis-related protein expression should also be measured for additional support.
7. APE patients often suffer from hypoxemia, with decreased PaO2 and increased PaCO2 being the main symptoms, a finding also mentioned in "Discussion" section of this manuscript. However, in "Results" section, it was found that R-hirudin increased PaO2 but had no significant effects on PaCO2 in rats, what was the reason for this? What did this phenomenon suggest? In addition, what was reason for the decrease in PaCO2 in APE rats at 4 d?

---

## Round 0.2 · accepted · Accept

The authors have addressed the comments from the reviewers.

·

Basic reporting

Authors have addressed my issues.

Experimental design

N/A

Validity of the findings

N/A

Reviewer 2 ·

Basic reporting

no comment

Experimental design

no comment

Validity of the findings

no comment

Reviewer 3 ·

Basic reporting

the revised manuscript is well written.

Experimental design

The experimental design is reasonable.

Validity of the findings

The findings is supported by the data.

Additional comments

None